# A CRITICAL ANALYSIS OF SELF-SUPERVISION, OR WHAT WE CAN LEARN FROM A SINGLE IMAGE

**Yuki M. Asano**     **Christian Rupprecht**     **Andrea Vedaldi**

Visual Geometry Group
University of Oxford
{yuki,chrisr,vedaldi}@robots.ox.ac.uk

## ABSTRACT

We look critically at popular self-supervision techniques for learning deep convolutional neural networks without manual labels. We show that three different and representative methods, BiGAN, RotNet and DeepCluster, can learn the first few layers of a convolutional network *from a single image* as well as using millions of images and manual labels, provided that strong data augmentation is used. However, for deeper layers the gap with manual supervision cannot be closed even if millions of unlabelled images are used for training. We conclude that: (1) the weights of the early layers of deep networks contain limited information about the statistics of natural images, that (2) such low-level statistics can be learned through self-supervision just as well as through strong supervision, and that (3) the low-level statistics can be captured via synthetic transformations instead of using a large image dataset.

## 1 INTRODUCTION

Despite tremendous progress in supervised learning, learning without external supervision remains difficult. Self-supervision has recently emerged as one of the most promising approaches to address this limitation. Self-supervision builds on the fact that convolutional neural networks (CNNs) transfer well between tasks (Shin et al., 2016; Oquab et al., 2014; Girshick, 2015; Huh et al., 2016). The idea then is to pre-train networks via *pretext tasks* that do not require expensive manual annotations and can be automatically generated from the data itself. Once pre-trained, networks can be applied to a target task by using only a modest amount of labelled data.

Early successes in self-supervision have encouraged authors to develop a large variety of pretext tasks, from colorization to rotation estimation and image autoencoding. Recent papers have shown performance competitive with supervised learning by learning complex neural networks on very large image datasets. Nevertheless, for a given model complexity, pre-training by using an off-the-shelf annotated image datasets such as ImageNet remains much more efficient.

In this paper, we aim to *investigate the effectiveness of current self-supervised approaches* by characterizing how much information they can extract from a given dataset of images. Since deep networks learn a *hierarchy* of representations, we further break down this investigation on a per-layer basis. We are motivated by the fact that the first few layers of most networks extract low-level information (Yosinski et al., 2014), and thus learning them may not require the high-level semantic information captured by manual labels.

Concretely, in this paper we answer the following simple question: "*is self-supervision able to exploit the information contained in a large number of images in order to learn different parts of a neural network?*"

We contribute two key findings. First, we show that *as little as a single image* is sufficient, when combined with self-supervision and data augmentation, to learn the first few layers of standard deep networks as well as using millions of images and full supervision (Figure 1). Hence, while self-supervised learning works well for these layers, this may be due more to the limited complexity of such features than the strength of the supervisory technique. This also confirms the intuition that early layers in a convolutional network amounts to low-level feature extractors, analogous to early

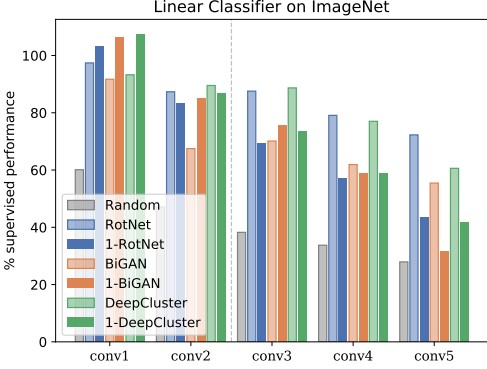

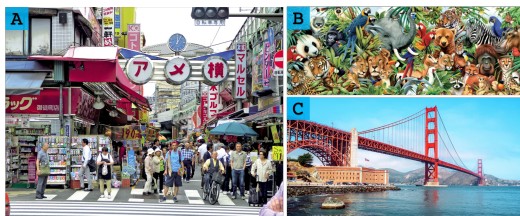

Figure 1: **Single-image self-supervision.** We show that several self-supervision methods can be used to train the first few layers of a deep neural networks using a *single training image*, such as this Image A, B or even C (above), provided that sufficient data augmentation is used.

learned and hand-crafted features for visual recognition (Olshausen & Field, 1997; Lowe, 2004; Dalal & Triggs, 2005). Finally, it demonstrates the importance of image transformations in learning such low-level features as opposed to image diversity.[1]

Our second finding is about the deeper layers of the network. For these, self-supervision remains inferior to strong supervision even if millions of images are used for training. Our finding is that this is unlikely to change with the addition of more data. In particular, we show that training these layers with self-supervision and a single image already achieves as much as two thirds of the performance that can be achieved by using a million different images.

We show that these conclusions hold true for three different self-supervised methods, BiGAN (Donahue et al., 2017), RotNet (Gidaris et al., 2018) and DeepCluster (Caron et al., 2018), which are representative of the spectrum of techniques that are currently popular. We find that performance as a function of the amount of data is dependent on the method, but all three methods can indeed leverage a single image to learn the first few layers of a deep network almost "perfectly".

Overall, while our results *do not* improve self-supervision *per-se*, they help to characterize the *limitations of current methods* and to better focus on the important open challenges.

## 2 RELATED WORK

Our paper relates to three broad areas of research: (a) self-supervised/unsupervised learning, (b) learning from a single sample, and (c) designing/learning low-level feature extractors. We discuss closely related work for each.

**Self-supervised learning:** A wide variety of proxy tasks, requiring no manual annotations, have been proposed for the self-training of deep convolutional neural networks. These methods use various cues and tasks namely, in-painting (Pathak et al., 2016), patch context and jigsaw puzzles (Doersch et al., 2015; Noroozi & Favaro, 2016; Noroozi et al., 2018; Mundhenk et al., 2017), clustering (Caron et al., 2018), noise-as-targets (Bojanowski & Joulin, 2017), colorization (Zhang et al., 2016; Larsson et al., 2017), generation (Jenni & Favaro, 2018; Ren & Lee, 2018; Donahue et al., 2017), geometry (Dosovitskiy et al., 2016; Gidaris et al., 2018) and counting (Noroozi et al., 2017). The idea is that the pretext task can be constructed automatically and easily on images alone. Thus, methods often modify information in the images and require the network to recover them. In-painting or colorization techniques fall in this category. However these methods have the downside that the features are learned on modified images which potentially harms the generalization to unmodified ones. For example, colorization uses a gray scale image as input, thus the network cannot learn to extract color information, which can be important for other tasks.

Slightly less related are methods that use additional information to learn features. Here, often temporal information is used in the form of videos. Typical pretext tasks are based on temporal-context (Misra et al., 2016; Wei et al., 2018; Lee et al., 2017; Sermanet et al., 2018), spatio-temporal

---

[1]Example applications that only rely on low-level feature extractors include template matching (Kat et al., 2018; Talmi et al., 2017) and style transfer (Gatys et al., 2016; Johnson et al., 2016), which currently rely on pre-training with millions of images.

cues (Isola et al., 2015; Gao et al., 2016; Wang et al., 2017), foreground-background segmentation via video segmentation (Pathak et al., 2017), optical-flow (Gan et al., 2018; Mahendran et al., 2018), future-frame synthesis (Srivastava et al., 2015), audio prediction from video (de Sa, 1994; Owens et al., 2016), audio-video alignment (Arandjelović & Zisserman, 2017), ego-motion estimation (Jayaraman & Grauman, 2015), slow feature analysis with higher order temporal coherence (Jayaraman & Grauman, 2016), transformation between frames (Agrawal et al., 2015) and patch tracking in videos (Wang & Gupta, 2015). Since we are interested in learning features from as little data as one image, we cannot make use of methods that rely on video input.

Our contribution inspects three unsupervised feature learning methods that use very different means of extracting information from the data: BiGAN (Donahue et al., 2017) utilizes a generative adversarial task, RotNet (Gidaris et al., 2018) exploits the photographic bias in the dataset and DeepCluster (Caron et al., 2018) learns stable feature representations under a number of image transformations by proxy labels obtained from clustering. These are described in more detail in the Methods section.

**Learning from a single sample:** In some applications of computer vision, the bold idea of learning from a single sample comes out of necessity. For general object tracking, methods such as max margin correlation filters (Rodriguez et al., 2013) learn robust tracking templates from a single sample of the patch. A single image can also be used to learn and interpolate multi-scale textures with a GAN framework (Rott Shaham et al., 2019). Single sample learning was pursued by the semi-parametric exemplar SVM model (Malisiewicz et al., 2011). They learn one SVM per positive sample separating it from all negative patches mined from the background. While only one sample is used for the positive set, the negative set consists of thousands of images and is a necessary component of their method. The negative space was approximated by a multi-dimensional Gaussian by the Exemplar LDA (Hariharan et al., 2012). These SVMs, one per positive sample, are pooled together using a max aggregation. We differ from both of these approaches in that we do not use a large collection of negative images to train our model. Instead we restrict ourselves to a single or a few images with a systematic augmentation strategy.

**Classical learned and hand-crafted low-level feature extractors:** Learning and hand-crafting features pre-dates modern deep learning approaches and self-supervision techniques. For example the classical work of (Olshausen & Field, 1997) shows that edge-like filters can be learned via sparse coding of just 10 natural scene images. SIFT (Lowe, 2004) and HOG (Dalal & Triggs, 2005) have been used extensively before the advent of convolutional neural networks and, in many ways, they resemble the first layers of these networks. The scatter transform of Bruna & Mallat (2013); Oyallon et al. (2017) is an handcrafted design that aims at replacing at least the first few layers of a deep network. While these results show that effective low-level features can be handcrafted, this is insufficient to clarify the power and limitation of self-supervision in deep networks. For instance, it is not obvious whether deep networks can learn better low level features than these, how many images may be required to learn them, and how effective self-supervision may be in doing so. For instance, as we also show in the experiments, replacing low-level layers in a convolutional networks with handcrafted features such as Oyallon et al. (2017) may still decrease the overall performance of the model. Furthermore, this says little about deeper layers, which we also investigate.

In this work we show that current deep learning methods learn slightly better low-level representations than hand crafted features such as the scattering transform. Additionally, these representations can be learned from one single image with augmentations and without supervision. The results show how current self-supervised learning approaches that use one million images yield only relatively small gains when compared to what can be achieved from one image and augmentations, and motivates a renewed focus on augmentations and incorporating prior knowledge into feature extractors.

## 3 METHODS

We discuss first our data and data augmentation strategy (section 3.1) and then we summarize the three different methods for unsupervised feature learning used in the experiments (section 3.2).

## 3.1 DATA

Our goal is to understand the performance of representation learning methods as a function of the image data used to train them. To make comparisons as fair as possible, we develop a protocol where only the nature of the training data is changed, but all other parameters remain fixed.

In order to do so, given a baseline method trained on $d$ source images, we replace those with another set of $d$ images. Of these, now only $N \ll d$ are *source* images (i.e. i.i.d. samples), while the remaining $d-N$ are augmentations of the source ones. Thus, the amount of information in the training data is controlled by $N$ and we can generate a continuum of datasets that vary from one extreme, utilizing a single source image $N=1$, to the other extreme, using all $N=d$ original training set images. For example, if the baseline method is trained on ImageNet, then $d=1{,}281{,}167$. When $N=1$, it means that we train the method using a single source image and generate the remaining $1{,}281{,}166$ images via augmentation. Other baselines use CIFAR-10/100 images, so in those cases $d=50{,}000$ instead.

The data augmentation protocol, is an extreme version of augmentations already employed by most deep learning protocols. Each method we test, in fact, already performs some data augmentation internally. Thus, when the method is applied on our augmented data, this can be equivalently thought of as incrementing these "native" augmentations by concatenating them with our own.

**Choice of augmentations.**   Next, we describe how the $N$ source images are expanded to additional $d-N$ images so that the models can be trained on exactly $d$ images, independent from the choice of $N$. The idea is to use an aggressive form of data augmentation involving cropping, scaling, rotation, contrast changes, and adding noise. These transformations are representative of invariances that one may wish to incorporate in the features. Augmentation can be seen as imposing a prior on how we expect the manifold of natural images to look like. When training with very few images, these priors become more important since the model cannot extract them directly from data.

Given a source image of size size $H \times W$, we first extract a certain number of random patches of size $(w,h)$, where $w \leq W$ and $h \leq H$ satisfy the additional constraints $\beta \leq \frac{wh}{WH}$ and $\gamma \leq \frac{h}{w} \leq \gamma^{-1}$. Thus, the smallest size of the crops is limited to be at least $\beta WH$ and at most the whole image. Additionally, changes to the aspect ratio are limited by $\gamma$. In practice we use $\beta = 10^{-3}$ and $\gamma = \frac{3}{4}$.

Second, good features should not change much by small image rotations, so images are rotated (before cropping to avoid border artifacts) by $\alpha \in (-35, 35)$ degrees. Due to symmetry in image statistics, images are also flipped left-to-right with 50% probability.

Illumination changes are common in natural images, we thus expect image features to be robust to color and contrast changes. Thus, we employ a set of linear transformations in RGB space to model this variability in real data. Additionally, the color/intensity of single pixels should not affect the feature representation, as this does not change the contents of the image. To this end, color jitter with additive brightness, contrast and saturation are sampled from three uniform distributions in $(0.6, 1.4)$ and hue noise from $(-0.1, 0.1)$ is applied to the image patches. Finally, the cropped and transformed patches are scaled to the color range $(-1, 1)$ and then rescaled to full $S \times S$ resolution to be supplied to each representation learning method, using bilinear interpolation. This formulation ensures that the patches are created in the target resolution $S$, independent from the size and aspect ratio $W, H$ of the source image.

**Real samples.**   The images used for the $N=1$ and $N=10$ experiments are shown in Figure 1 and the appendix respectively (this is *all* the training data used in such experiments). For the special case of using a single training image, i.e. $N=1$, we have chosen one photographic ($2560 \times 1920$) and one drawn image ($600 \times 225$), which we call *Image A* and *Image B*, respectively. The two images were manually selected as they contain rich texture and are diverse, but their choice was not optimized for performance. We test only two images due to the cost of running a full set of experiments (each image is expanded up to 1.2M times for training some of the models, as explained above). However, this is sufficient to prove our main points. We also test another ($1165 \times 585$) *Image C* to ablate the "crowdedness" of an image, as this latter contains large areas covering no objects. While resolution matters to some extent as a bigger image contains more pixels, the information within is still far more correlated, and thus more redundant than sampling several smaller images. In particular, the resolution difference in Image A and B appears to be negligible in our experiments. For CIFAR-10, where $S=32$ we only use Image B due to the resolution difference. In direct comparison, Image B

is the size of about 132 CIFAR images which is still much less than $d = 50,000$. For $N > 1$, we select the source images randomly from each method's training set.

## 3.2 REPRESENTATION LEARNING METHODS

**Generative models.** Generative Adversarial Networks (GANs) (Goodfellow et al., 2014) learn to generate images using an adversarial objective: a generator network maps noise samples to image samples, approximating a target image distribution and a discriminator network is tasked with distinguishing generated and real samples. Generator and discriminator are pitched one against the other and learned together; when an equilibrium is reached, the generator produces images indistinguishable (at least from the viewpoint of the discriminator) from real ones.

Bidirectional Generative Adversarial Networks (BiGAN) (Donahue et al., 2017; Dumoulin et al., 2016) are an extension of GANs designed to learn a useful image representation as an approximate inverse of the generator through joint inference on an encoding and the image. This method's native augmentation uses random crops and random horizontal flips to learn features from $S = 128$ sized images. As opposed to the other two methods discussed below it employs leaky ReLU non-linearities as is typical in GAN discriminators.

**Rotation.** Most image datasets contain pictures that are 'upright' as this is how humans prefer to take and look at them. This photographer bias can be understood as a form of implicit data labelling. RotNet (Gidaris et al., 2018) exploits this by tasking a network with predicting the upright direction of a picture after applying to it a random rotation multiple of 90 degrees (in practice this is formulated as a 4-way classification problem). The authors reason that the concept of 'upright' requires learning high level concepts in the image and hence this method is not vulnerable to exploiting low-level visual information, encouraging the network to learn more abstract features. In our experiments, we test this hypothesis by learning from impoverished datasets that may lack the photographer bias. The native augmentations that RotNet uses on the $S = 256$ inputs only comprise horizontal flips and non-scaled random crops to $224 \times 224$.

**Clustering.** DeepCluster (Caron et al., 2018) is a recent state-of-the-art unsupervised representation learning method. This approach alternates $k$-means clustering to produce pseudo-labels for the data and feature learning to fit the representation to these labels. The authors attribute the success of the method to the prior knowledge ingrained in the structure of the convolutional neural network (Ulyanov et al., 2018).

The method alternatives between a clustering step, in which $k$-means is applied on the PCA-reduced features with $k = 10^4$, and a learning step, in which the network is trained to predict the cluster ID for each image under a set of augmentations (random resized crops with $\beta = 0.08, \gamma = \frac{3}{4}$ and horizontal flips) that constitute its native augmentations used on top of the $S = 256$ input images.

## 4 EXPERIMENTS

We evaluate the representation learning methods on ImageNet and CIFAR-10/100 using linear probes (Section 4.1). After ablating various choices of transformations in our augmentation protocol (Section 4.2), we move to the core question of the paper: whether a large dataset is beneficial to unsupervised learning, especially for learning early convolutional features (Section 4.3).

## 4.1 LINEAR PROBES AND BASELINE ARCHITECTURE

In order to quantify if a neural network has learned useful feature representations, we follow the standard approach of using linear probes (Zhang et al., 2017). This amounts to solving a difficult task such as ImageNet classification by training a linear classifier on top of pre-trained feature representations, which are kept fixed. Linear classifiers heavily rely on the quality of the representation since their discriminative power is low.

We apply linear probes to all intermediate convolutional layers of networks and train on the ImageNet LSVRC-12 (Deng et al., 2009) and CIFAR-10/100 (Krizhevsky, 2009) datasets, which are the standard benchmarks for evaluation in self-supervised learning. Our base encoder architecture is AlexNet (Krizhevsky et al., 2012) with BatchNorm, since this is a good representative model and is most often used in other unsupervised learning work for the purpose of benchmarking. This model

|     |              | CIFAR-10 | | | |
| --- | ------------ | ----- | ----- | ----- | ----- |
|     |              | conv1 | conv2 | conv3 | conv4 |
| (a) | Fully sup.   | 66.5  | 70.1  | 72.4  | 75.9  |
| (b) | Random feat. | 57.8  | 55.5  | 54.2  | 47.3  |
| (c) | No aug.      | 57.9  | 56.2  | 54.2  | 47.8  |
| (d) | Jitter       | 58.9  | 58.0  | 57.0  | 49.8  |
| (e) | Rotation     | 61.4  | 58.8  | 56.1  | 47.5  |
| (f) | Scale        | 67.9  | 69.3  | 67.9  | 59.1  |
| (g) | Rot. & jitter | 64.9 | 63.6  | 61.0  | 53.4  |
| (h) | Rot. & scale | 67.6  | 69.9  | 68.0  | 60.7  |
| (i) | Jitter & scale | 68.1 | 71.3 | 69.5  | 62.4  |
| (j) | All          | **68.1** | **72.3** | **70.8** | **63.5** |

Table 1: **Ablating data augmentation using MonoGAN (left).** Training a linear classifier on the features extracted at different depths of the network for CIFAR-10.

Table 2: **ImageNet LSVRC-12 linear probing evaluation (below).** A linear classifier is trained on the (downsampled) activations of each layer in the pretrained model. We report classification accuracy averaged over 10 crops. The ‡ indicated that numbers are taken from (Zhang et al., 2017).

|     |                                       |          | ILSVRC-12 | | | | |
| --- | ------------------------------------- | -------- | ----- | ----- | ----- | ----- | ----- |
|     | Method, Reference                     | #images  | conv1 | conv2 | conv3 | conv4 | conv5 |
| (a) | Full-supervision‡                     | 1,281,167 | 19.3 | 36.3  | 44.2  | 48.3  | 50.5  |
| (b) | (Oyallon et al., 2017): Scattering    | 0        | -     | 18.9  | -     | -     | -     |
| (c) | Random‡                               | 0        | 11.6  | 17.1  | 16.9  | 16.3  | 14.1  |
| (d) | (Krähenbühl et al., 2016):$k$-means‡  | ≈160     | 17.5  | 23.0  | 24.5  | 23.2  | 20.6  |
| (e) | (Donahue et al., 2017): BiGAN‡        | 1,281,167 | 17.7 | 24.5  | 31.0  | 29.9  | 28.0  |
| (f) | mono, Image A                         | 1        | 20.4  | 30.9  | 33.4  | 28.4  | 16.0  |
| (g) | mono, Image B                         | 1        | 20.5  | 30.4  | 31.6  | 27.0  | 16.8  |
| (h) | deka                                  | 10       | 16.2  | 16.5  | 16.5  | 13.1  | 7.5   |
| (i) | kilo                                  | 1,000    | 16.1  | 17.7  | 18.3  | 17.6  | 13.5  |
| (j) | (Gidaris et al., 2018): RotNet        | 1,281,167 | 18.8 | 31.7  | 38.7  | 38.2  | 36.5  |
| (k) | mono, Image A                         | 1        | 19.9  | 30.2  | 30.6  | 27.6  | 21.9  |
| (l) | mono, Image B                         | 1        | 17.8  | 27.6  | 27.9  | 25.4  | 20.2  |
| (m) | deka                                  | 10       | 19.6  | 30.7  | 32.6  | 28.9  | 22.6  |
| (n) | kilo                                  | 1,000    | 21.0  | 33.5  | 36.5  | 34.0  | 29.4  |
| (o) | (Caron et al., 2018): DeepCluster     | 1,281,167 | 18.0 | 32.5  | 39.2  | 37.2  | 30.6  |
| (p) | mono, Image A                         | 1        | 20.7  | 31.5  | 32.5  | 28.5  | 21.0  |
| (q) | mono, Image B                         | 1        | 19.7  | 30.1  | 31.6  | 28.5  | 20.4  |
| (r) | mono, Image C                         | 1        | 18.9  | 29.2  | 31.5  | 28.9  | 23.5  |
| (s) | deka                                  | 10       | 18.5  | 29.0  | 31.1  | 28.2  | 21.9  |
| (t) | kilo                                  | 1,000    | 19.5  | 29.8  | 33.0  | 31.7  | 26.8  |

Table 3: **CIFAR-10/100.** Accuracy of linear classifiers on different network layers.

| Dataset |  | CIFAR-10 | | | | CIFAR-100 | | | |
| --- | --- | --- | --- | --- | --- | --- | --- | --- | --- |
| Model |  | conv1 | conv2 | conv3 | conv4 | conv1 | conv2 | conv3 | conv4 |
| Fully supervised | | 66.5 | 70.1 | 72.4 | 75.9 | 38.7 | 43.6 | 44.4 | 46.5 |
| Random | | 57.8 | 55.5 | 54.2 | 47.3 | 30.9 | 29.8 | 28.6 | 24.1 |
| RotNet | | 64.4 | 65.6 | 65.6 | 59.1 | 36.0 | 35.9 | 34.2 | 25.8 |
| GAN (CIFAR-10) | | 67.7 | **73.0** | **72.5** | **69.2** | 39.6 | 46.0 | **45.1** | 39.9 |
| GAN (CIFAR-100) | | - | - | - | - | 38.1 | 42.2 | 44.0 | **46.6** |
| MonoGAN | | **68.1** | 72.3 | 70.8 | 63.5 | **39.9** | **46.9** | 44.5 | 38.8 |

has five convolutional blocks (each comprising a linear convolution later followed by ReLU and optionally max pooling). We insert the probes right after the ReLU layer in each block, and denote these entry points `conv1` to `conv5`. Applying the linear probes at each convolutional layer allows studying the quality of the representation learned at different depths of the network.

**Details.** While linear probes are conceptually straightforward, there are several technical details that affect the final accuracy by a few percentage points. Unfortunately, prior work has used several slightly different setups, so that comparing results of different publications must be done with caution. To make matters more difficult, not all papers released evaluation source code. We prove this standardized testing code here[2].

---

[2] https://github.com/yukimasano/linear-probes

In our implementation, we follow the original proposal (Zhang et al., 2017) in pooling each representation to a vector with $9600, 9216, 9600, 9600, 9216$ dimensions for `conv1-5` using adaptive max-pooling, and absorb the batch normalization weights into the preceding convolutions. For evaluation on ImageNet we follow RotNet to train linear probes: images are resized such that the shorter edge has a length of 256 pixels, random crops of $224 \times 224$ are computed and flipped horizontally with $50\%$ probability. Learning lasts for 36 epochs and the learning rate schedule starts from $0.01$ and is divided by five at epochs 5, 15 and 25. The top-1 accuracy of the linear classifier is then measured on the ImageNet validation subset. This uses DeepCluster's protocol, extracting 10 crops for each validation image (four at the corners and one at the center along with their horizontal flips) and averaging the prediction scores before the accuracy is computed. For CIFAR-10/100 data, we follow the same learning rate schedule and for both training and evaluation we do not reduce the dimensionality of the representations and keep the images' original size of $32 \times 32$.

## 4.2 EFFECT OF AUGMENTATIONS

In order to better understand which image transformations are important to learn a good feature representations, we analyze the impact of augmentation settings. For speed, these experiments are conducted using the CIFAR-10 images ($d = 50,000$ in the training set) and with the smaller source Image B and a GAN using the Wasserstein GAN formulation with gradient penalty (Gulrajani et al., 2017). The encoder is a smaller AlexNet-like CNN consisting of four convolutional layers (kernel sizes: $7, 5, 3, 3$; strides: $3, 2, 2, 1$) followed by a single fully connected layer as the discriminator. Given that the GAN is trained on a single image (w/ augmentations), we call this setting *MonoGAN*.

Table 1 reports all $2^3$ combinations of the three main augmentations (scale, rotation, and jitter) and a randomly initialized network baseline (see Table 1 (b)) using the linear probes protocol discussed above. Without data augmentation the model only achieves marginally better performance than the random network (which also achieves a non-negligible level of performance (Ulyanov et al., 2017; Caron et al., 2018)). This is understandable since the dataset literally consists of a single training image cloned $d$ times. Color jitter and rotation slightly improve the performance of all probes by 1-2% points, but random rescaling adds at least ten points at every depth (see Table 1 (f,h,i)) and is the most important single augmentation. A similar conclusion can be drawn when two augmentations are combined, although there are diminishing returns as more augmentations are combined. Overall, we find all three types of augmentations are of importance when training in the ultra-low data setting.

## 4.3 BENCHMARK EVALUATION

We analyze how performance varies as a function $N$, the number of actual samples that are used to generated the augmented datasets, and compare it to the gold-standard setup (in terms of choice of training data) defined in the papers that introduced each method. The evaluation is again based on linear probes (Section 4.1).

**Mono is enough.** From Table 2 we make the following observations. Training with just a single source image (f,g,k,l,p,q) is much better than random initialization (c) for all layers. Notably, these models also outperform Gabor-like filters from Scattering networks (Bruna & Mallat, 2013), which are hand crafted image features, replacing the first two convolutional layers as in (Oyallon et al., 2017). Using the same protocol as in the paper, this only achieves an accuracy of $18.9\%$ compared to (p)'s `conv2` $> 30\%$.

More importantly, when comparing within pretext task, even with one image we are able to improve the quality of `conv1-conv3` features compared to full (unsupervised) ImageNet training for GAN based self-supervision (e-i). For the other methods (j-n, o-s) we reach and also surpass the performance for the first layer and are within $1.5\%$ points for the second. Given that the best unsupervised performance for `conv2` is 32.5, our method using a single source Image A (Table 2, p) is remarkably close with 31.5.

**Image contents.** While we surpass the GAN based approach of (Donahue et al., 2017) for both single source images, we find more nuanced results for the other two methods: For RotNet, as expected, the photographic bias cannot be extracted from a single image. Thus its performance is low with little training data and increases together with the number of images (Table 2, j-n). When comparing Image A and B trained networks for RotNet, we find that the photograph yields better performance than the hand drawn animal image. This indicates that the method can extract rotation

| Method, Image A | | | Method, Image B | | |
| --- | --- | --- | --- | --- | --- |
| BiGAN | RotNet | DeepCluster | BiGAN | RotNet | DeepCluster |
| 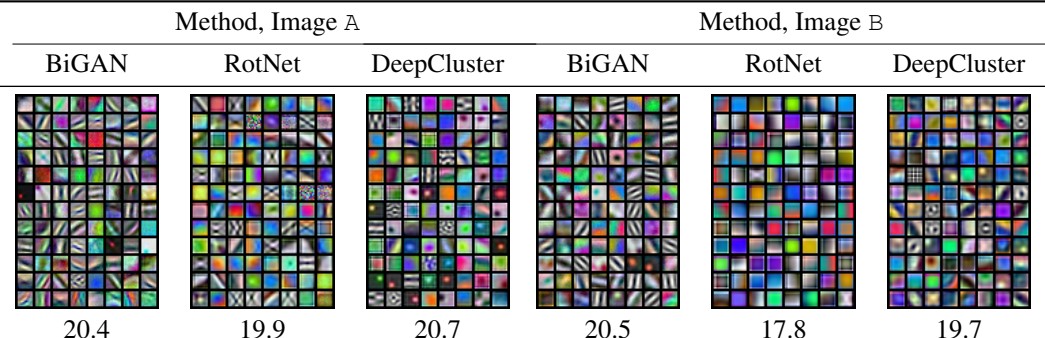 | | | | | |
| 20.4 | 19.9 | 20.7 | 20.5 | 17.8 | 19.7 |

Figure 2: **`conv1` filters trained using a single image.** The 96 learned $(3 \times 11 \times 11)$ filters for the first layer of AlexNet are shown for each single training image and method along with their linear classifier performance. For visualization, each filter is normalized to be in the range of $(-1, 1)$.

information from low level image features such as patches which is at first counter intuitive. Considering that the hand-drawn image does not work well, we can assume that lighting and shadows even in small patches can indeed give important cues on the up direction which can be learned even from a single (real) image. DeepCluster shows poor performance in `conv1` which we can improve upon in the single image setting (Table 2, o-r). Naturally, the image content matters: a trivial image without any image gradient (e.g. picture of a white wall) would not provide enough signal for any method. To better understand this issue, we also train DeepCluster on the much less cluttered Image C to analyze how much the image influences our claims. We find that even though this image contains large parts of sky and sea, the performance is only slightly lower than that of Image A. This finding indicates that the augmentations can even compensate for large untextured areas and the exact choice of image is not critical.

**More than one image.** While BiGAN fails to converge for $N \in \{10, 1000\}$, most likely due to issues in learning from a distribution which is neither whole images nor only patches, we find that both RotNet and DeepCluster improve their performance in deeper layers when increasing the number of training images. However, for `conv1` and `conv2`, a single image is enough. In deeper layers, DeepCluster seems to require large amounts of source images to yield the reported results as the deka- and kilo- variants start improving over the single image case (Table 2, o-t). This need for data also explains the gap between the two input images which have different resolutions. Summarizing Table 2, we can conclude that learning `conv1`, `conv2` and for the most part `conv3` (33.4 vs. 39.4) on over 1M images does not yield a significant performance increase over using one single training image — a highly unexpected result.

**Generalization.** In Table 3, we show the results of training linear classifiers for the CIFAR-10 dataset and compare against various baselines. We find that the GAN trained on the smaller Image B outperforms all other methods including the fully-supervised trained one for the first convolutional layer. We also outperform the same architecture trained on the full CIFAR-10 training set using RotNet, which might be due to the fact that either CIFAR images do not contain much information about the orientation of the picture or because they do not contain as many objects as in ImageNet. While the GAN trained on the whole dataset outperforms the MonoGAN on the deeper layers, the gap stays very small until the last layer. These findings are also reflected in the experiments on the CIFAR-100 dataset shown in Table 3. We find that our method obtains the best performance for the first two layers, even against the fully supervised version. The gap between our mono variant and the other methods increases again with deeper layers, hinting to the fact that we cannot learn very high level concepts in deeper layers from just one single image. These results corroborate the finding that our method allows learning very generalizable early features that are not domain dependent.

## 4.4 QUALITATIVE ANALYSIS

**Visual comparison of weights.** In Figure 2, we compare the learned filters of all first-layer convolutions of an AlexNet trained with the different methods and a single image. First, we find that the filters closely resemble those obtained via supervised training: Gabor-like edge detectors and various color blobs. Second, we find that the look is not easily predictive of its performance, e.g.

while generatively learned filters (BiGAN) show many edge detectors, its linear probes performance is about the same as that of DeepCluster which seems to learn many somewhat redundant point features. However, we also find that some edge detectors are required, as we can confirm from RotNet and DeepCluster trained on Image B, which yield less crisp filters and worse performances.

Table 4: **Finetuning experiments** The pretrained model's first two convolutions are left frozen (or replaced by the Scattering transform) and the nework is retrained using ImageNet LSVRC-12 training set.

|  | Top-1 |
|---|---|
| Full sup. | 59.4 |
| Random | 42.6 |
| Scattering | 49.2 |
| BiGAN, A | 51.4 |
| RotNet, A | 49.5 |
| DeepCluster A | 52.5 |

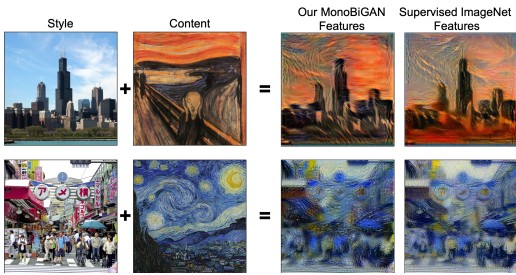

Figure 3: **Style transfer with single-image pretraining.** We show two style transfer results using the Image A trained BiGAN and the ImageNet pretrained AlexNet.

**Fine-tuning instead of freezing.**  In Tab. 4, we show the results of retraining a network with the first two convolutional filters, or the scattering transform from (Oyallon et al., 2017), left frozen. We observe that our single image trained DeepCluster and BiGAN models achieve performances closes to the supervised benchmark. Notably, the scattering transform as a replacement for conv1-2 performs slightly worse than the analyzed single image methods. We also show in the appendix the results of retraining a network initialized with the first two convolutional layers obtained from a single image and subsequently linearly probing the model. The results are shown in Appendix Tab. 5 and we find that we can recover the performance of fully-supervised networks, i.e. the first two convolutional filters trained from just a single image generalize well and do not get stuck in an image specific minimum.

**Neural style transfer.**  Lastly, we show how our features trained on only a single image can be used for other applications. In Figure 3 we show two basic style transfers using the method of (Gatys et al., 2016) from an official PyTorch tutorial[3]. Image content and style are separated and the style is transferred from the source to target image using all CNN features, not just the shallow layers. We visually compare the results of using our features and from full ImageNet supervision. We find almost no visual differences in the stylized images and can conclude that our early features are equally powerful as fully supervised ones for this task.

## 5 CONCLUSIONS

We have made the surprising observation that we can learn good and generalizable features through self-supervision from one single source image, provided that sufficient data augmentation is used. Our results complement recent works (Mahajan et al., 2018; Goyal et al., 2019) that have investigated self-supervision in the very large data regime. Our main conclusion is that these methods succeed perfectly in capturing the simplest image statistics, but that for deeper layers a gap exist with strong supervision which is compensated only in limited manner by using large datasets. This novel finding motivates a renewed focus on the role of augmentations in self-supervised learning and critical rethinking of how to better leverage the available data.

ACKNOWLEDGEMENTS.

We thank Aravindh Mahendran for fruitful discussions. Yuki Asano gratefully acknowledges support from the EPSRC Centre for Doctoral Training in Autonomous Intelligent Machines & Systems (EP/L015897/1). The work is supported by ERC IDIU-638009.

---

[3]https://pytorch.org/tutorials/advanced/neural_style_tutorial.html

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

# A    APPENDIX

## A.1    IMAGENET TRAINING IMAGES

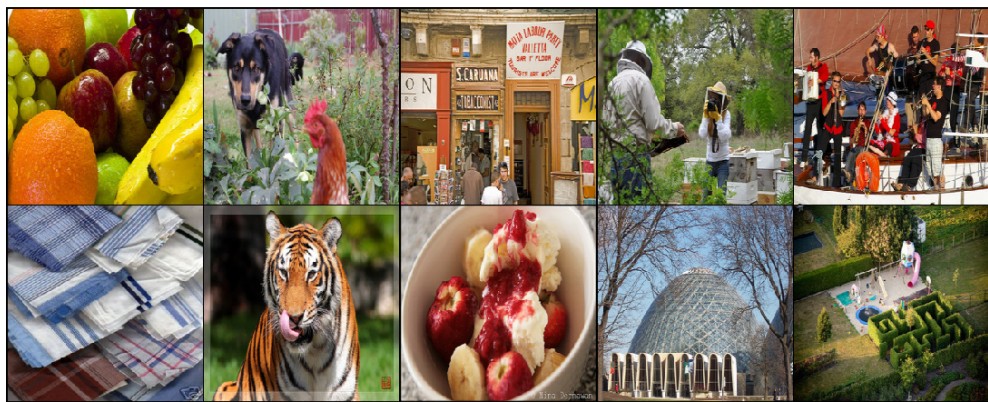

Figure 4: ImageNet images for the $N = 10$ experiments.

The images used for the $N = 10$ experiments are shown in fig. 4.

## A.2    VISUAL COMPARISON OF FILTERS

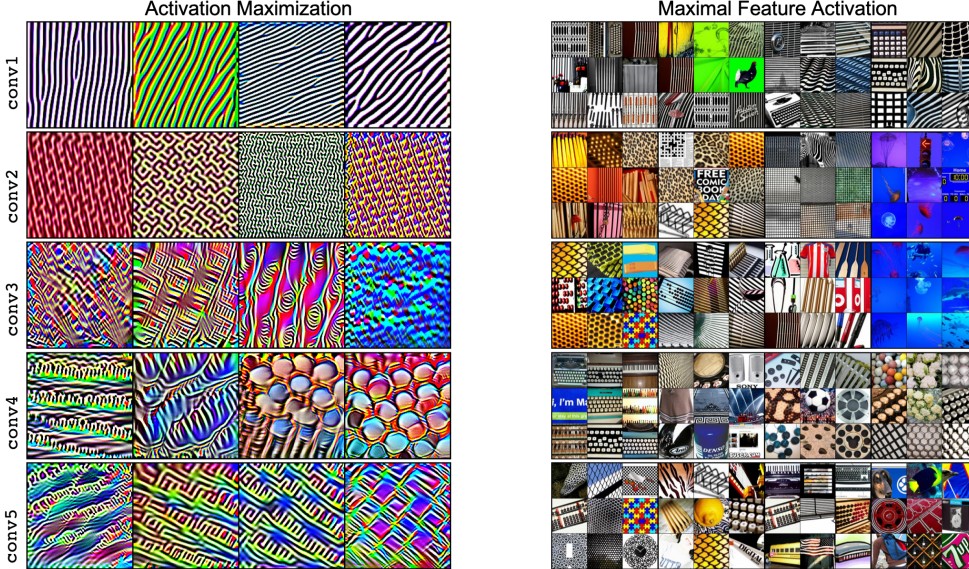

Figure 5: **Filter visualization.** We show activation maximization (left) and retrieval of top 9 activated images from the training set of ImageNet (right) for four random non-cherrypicked target filters. From top to bottom: `conv1-5` of the BiGAN trained on a single image `A`. The filter visualization is obtained by learning a (regularized) input image that maximizes the response to the target filter using the library Lucid (Olah et al., 2018).

In order to understand what deeper neurons are responding to in our model, we visualize random neurons via activation maximization (Erhan et al., 2009; Zeiler & Fergus, 2014) in each layer. Additionally, we retrieve the top-9 images in the ImageNet training set that activate each neuron most in Figure 5. Since the mono networks are not trained on the ImageNet dataset, it can be used here for visualization. From the first convolutional layer we find typical neurons strongly reacting to oriented edges. In layers 2-4 we find patterns such as grids (`conv2:3`), and textures such as leopard skin (`conv2:2`) and round grid cover (`conv4:4`). Confirming our hypothesis that the neural network is only extracting patterns and not semantic information, we do not find any neurons particularly specialized to certain objects even in higher levels as for example dog faces or similar which can be fund in supervised networks. This finding aligns with the observations of other unsupervised methods (Caron et al., 2018; Zhang et al., 2017). As most neurons extract simple patterns and

Table 5: **Finetuning experiments** Models are initialized using `conv1` and `conv2` from various single image trained models and the *whole* network is fine-tuned using ImageNet LSVRC-12 training set. Accuracy is averaged over 10 crops.

|  | c1 | c2 | c3 | c4 | c5 |
|---|---|---|---|---|---|
| Full sup. | 19.3 | 36.3 | 44.2 | 48.3 | 50.5 |
| BiGAN, A | 22.5 | 37.6 | 44.2 | 47.6 | 48.3 |
| RotNet, A | 22.0 | 38.2 | 44.8 | 49.2 | 51.8 |
| DeepCluster, A | 21.8 | 35.9 | 43.6 | 48.8 | 50.4 |

textures, the surprising effectiveness of training a network using a single image can be explained by the recent finding that even CNNs trained on ImageNet rely on texture (as opposed to shape) information to classify (Geirhos et al., 2019).

### A.3 RETRAINING FROM SINGLE IMAGE INITIALIZATION

In Table 5, we initialize AlexNet models using the first two convolutional filters learned from a single image and retrain them using ImageNet. We find that the networks recover their performance fully and the first filters do not make the network stuck in a bad local minimum despite having been trained on a single image from a different distribution. The difference from the BiGAN to the full supervision model is likely due to it using a smaller input resolution (112 instead of 224), as the BiGAN's output resolution is limited.

### A.4 LINEAR PROBES ON IMAGENET

We show two plots of the ImageNet linear probes results (Table 2 of the paper) in fig. 6. On the left we plot performance per layer in absolute scale. Naturally the performance of the supervised model improves with depth, while all unsupervised models degrade after `conv3`. From the relative plot on the right, it becomes clear that with our training scheme, we can even slightly surpass supervised performance on `conv1` presumably since our model is trained with sometimes very small patches, thus receiving an emphasis on learning good low level filters. The gap between all self-supervised methods and the supervised baseline increases with depth, due to the fact that the supervised model is trained for this specific task, whereas the self-supervised models learn from a surrogate task without labels.

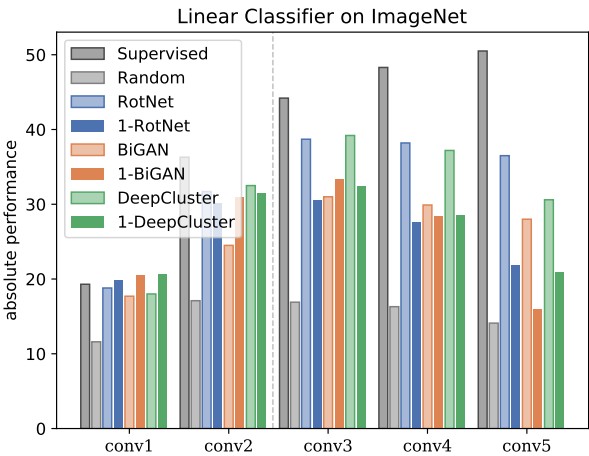

Figure 6: **Linear Classifiers on ImageNet.** Classification accuracies of linear classifiers trained on the representations from Table 2 are shown in absolute scale.

### A.5 EXAMPLE AUGMENTED TRAINING DATA

In figs. 7 to 10 we show example patches generated by our augmentation strategy for the datasets with different N. Even though the images and patches are very different in color and shape distribu-

tion, our model learns weights that perform similarly in the linear probes benchmark (see Table 2 in the paper).

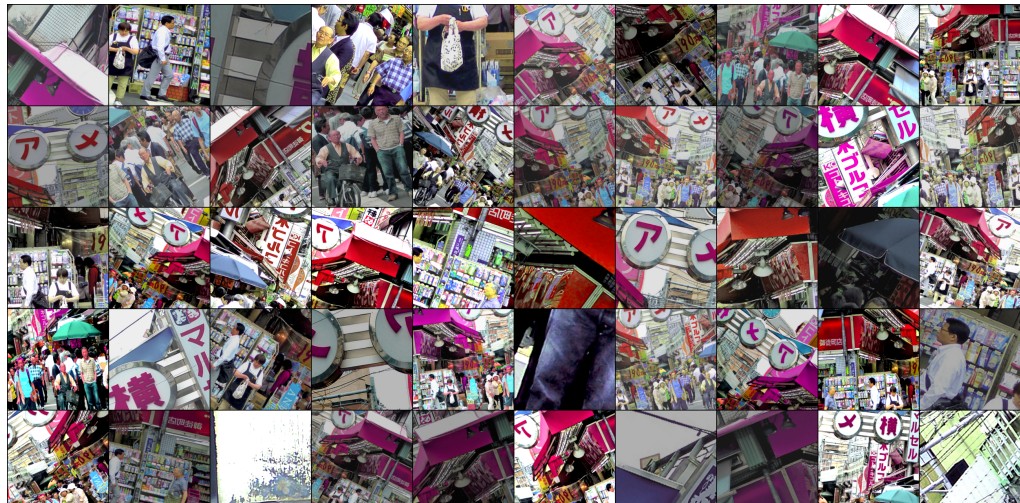

Figure 7: **Example crops of Image A** ($N = 1$) **dataset.**

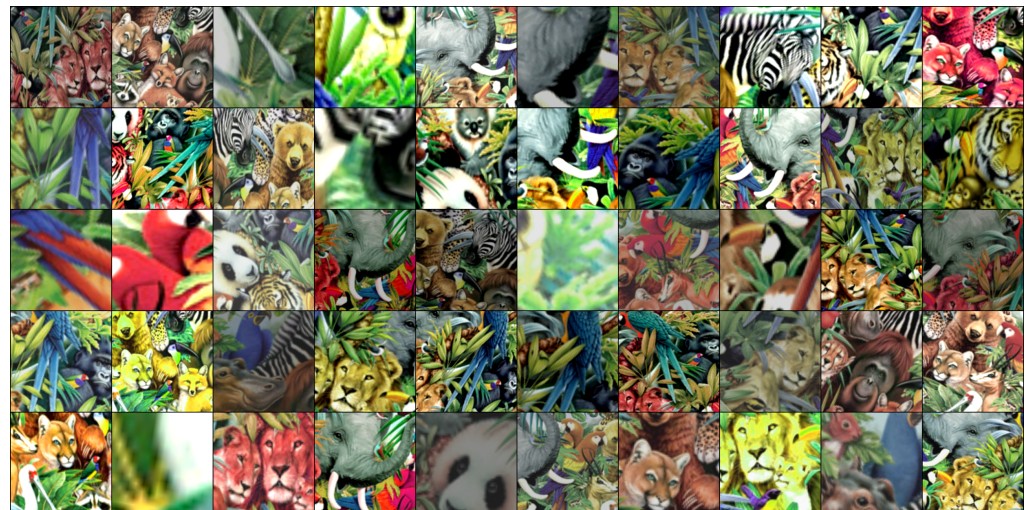

Figure 8: **Example crops of Image B** ($N = 1$) **dataset.** 50 samples were selected randomly.

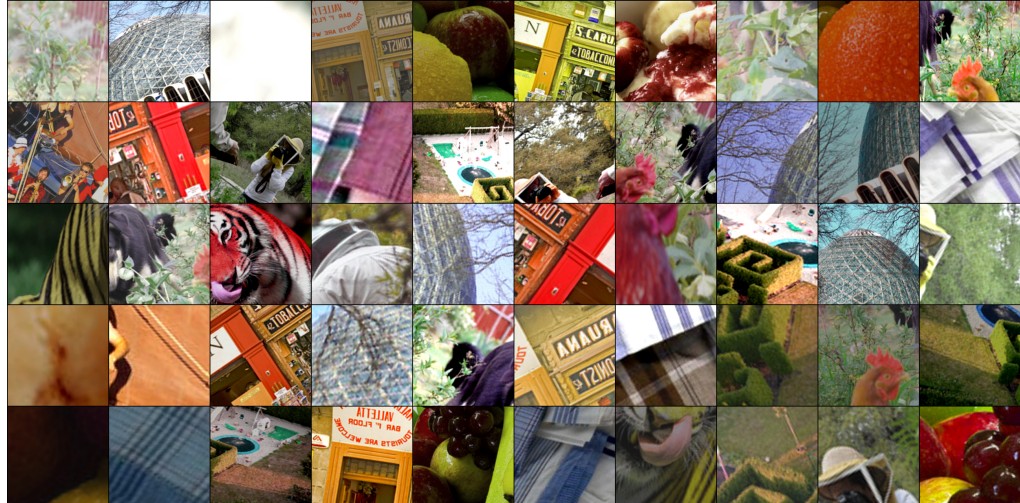

Figure 9: **Example crops of deka** ($N = 10$) **dataset.** 50 samples were selected randomly.

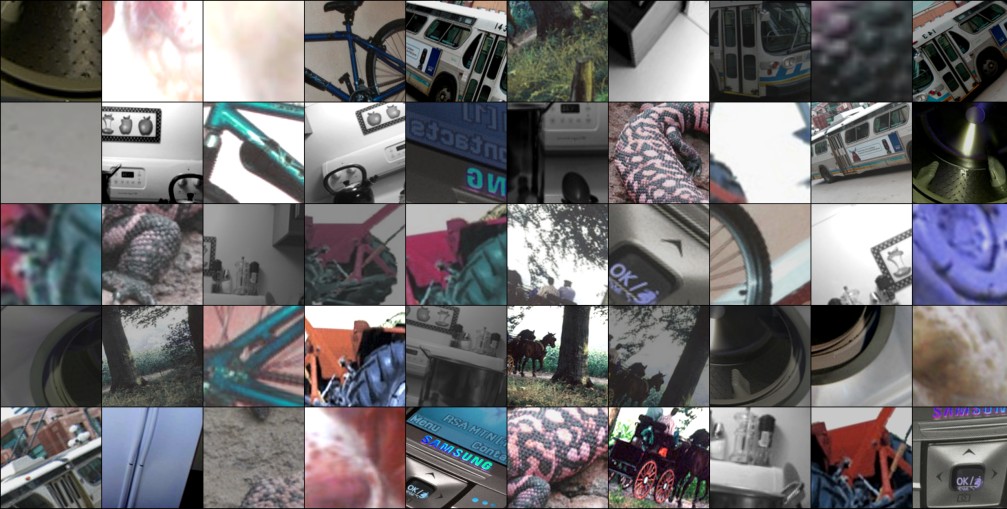

Figure 10: **Example crops of kilo** ($N = 1000$) **dataset.** 50 samples were selected randomly.

