# OpenReview forum: "A critical analysis of self-supervision, or what we can learn from a single image"
_ICLR.cc/2020/Conference — Accept (Poster)_

### Official Review · AnonReviewer3 · 2019-10-23
**Official Blind Review #3**

**Rating:** 6

**Review:**

The paper studies self-supervised learning from very few unlabeled images, down to the extreme case where only a single image is used for training. From the few/single image(s) available for training, a data set of the same size as some unmodified reference data set (ImageNet, Cifar-10/100) is generated through heavy data augmentation (cropping, scaling, rotation, contrast changes, adding noise). Three popular self-supervised learning algorithms are then trained on this data sets, namely (Bi)GAN, RotNet, and DeepCluster, and the linear probing accuracy on different blocks is compared to that obtained by training the same methods on the reference data sets. The linear probing accuracy from the first few conv layers of the network trained on the single/few image data set is found to be comparable to or better than that of the same model trained on the full reference data set.

I enjoyed the paper; it addresses the interesting setting of an extremely small data set which complements the large number of studies on scaling up self-supervised learning algorithms. I think it is not extremely surprising that using the proposed strategy allows to learn low level features as captured by the first few layers, but I think it is worth studying and quantifying. The experiments are carefully described and presented, and the paper is well-written.

Here are a few questions and concerns:

- How much does the image matter for the single-image data set? The selected images A and B are of very high entropy and show a lot of different objects (image A) and animals (image B). How do the results change if e.g. a landscape image or an abstract architecture photo is used?

- How general is the proposed approach? How likely is it to generalize to other approaches such as Jigsaw (Doersch et al., 2015) and Exemplar (Dosovitskiy et al., 2016)? It would be good to comment on this.

- [1] found that the network architecture for self-supervised learning can matter a lot, and that by using a ResNet architecture, performance of SSL methods can be significantly improved. In particular, the linear probing accuracy appears to be often monotonic as a function of the depth of the layer it is computed from. This is in contrast to what is observed for AlexNet in Tables 2 and 3, where the conv5 accuracy is lower than the conv4. It would therefore be instructive to add experiments for ResNet to see how well the results generalize to other network architectures.

- Does the MonoGAN exhibit stable training dynamics comparable to training WGAN on CIFAR-10, or do the training dynamics change on the single-image data set?


Overall, I’m leaning towards accepting the paper, but it would be important to see how well the experiments generalize to i) ResNet and ii) other (lower entropy) input images.

[1] Kolesnikov, A., Zhai, X. and Beyer, L., 2019. Revisiting self-supervised visual representation learning. arXiv preprint arXiv:1901.09005.



---
Update after rebuttal:
I thank the authors for their detailed response. I appreciate the efforts of the authors into investigating the issues raised, the described experiments sound promising. Unfortunately, the new results are not presented in the revision. I will therefore keep my rating.

**Experience Assessment:**

I have published one or two papers in this area.

**Review Assessment: Checking Correctness Of Derivations And Theory:**

I assessed the sensibility of the derivations and theory.

**Review Assessment: Checking Correctness Of Experiments:**

I carefully checked the experiments.

**Review Assessment: Thoroughness In Paper Reading:**

I read the paper at least twice and used my best judgement in assessing the paper.

---

> ### Author Response · Authors · 2019-11-07
> **Response to Review #3**
>
> We thank the reviewer for their time and their clear understanding of the key aspects of the paper. We address the reviewer’s questions in the following:
>
> >  How much does the image matter for the single-image data set?
>
> The reviewer raises an important point about the tested single images. Less crowded images could lead to many patches having no gradients (e.g. showing only the sky), leading to a failure of at least RotNet, if not also BiGAN on many samples of the augmented dataset. Our image choices were thus motivated by striving for simplicity and not further adding a pipeline that would, for example, extract only patches with sufficiently large image gradients. We are training DeepCluster now on a significantly less busy image and will report results in the coming days.
>
>
> >  How general is the proposed approach?
>
> We believe that this method will work well for pretext tasks that rely on learning via detecting and learning invariances, such as Exemplar [1], Colorization [2], and Noise-as-targets [3]. Methods such as Context [4] and Jigsaw [5] could potentially work less well as they would potentially easily find a way to cheat given the limited amount of original data of one image. However, as the authors note in the paper cited by the reviewer, the accuracy of a pretext task does not translate to downstream task performances, so even a method that is simple on one image’s patches does not necessarily fail.
> This is an interesting avenue for research and we hope that this paper could inspire follow-up work on this topic.
>
>
> > [1] found that the network architecture for self-supervised learning can matter a lot, and that by using a ResNet architecture, performance of SSL methods can be significantly improved.
>
> Indeed, the paper mentioned by the reviewer shows that the performance of various self-supervised methods for ResNets does not degrade with the depth as it does for VGG and AlexNets due to the skip-connections. However, as ResNets have not been originally used to train the methods analyzed in our paper, we have stayed in the bounds that are required for fair comparisons and only used AlexNet. We agree with the reviewer that it would be good to check if ResNets, in general, can also be trained in such a manner (e.g. could global pooling destroy the signal?), so we are running an experiment on a ResNet-18 and will report results in the upcoming days.
>
>
> > Does the MonoGAN exhibit stable training dynamics comparable to training WGAN on CIFAR-10, or do the training dynamics change on the single-image data set?
>
> MonoGAN trained without any exploding gradients or other problems frequently encountered by GANs. As we have suggested in the paper, this might be due to the fact that image-patches from one image follow a simpler distribution than in-the-wild images of a complete dataset.
>
> —
> [1] A. Dosovitskiy et al. "Discriminative unsupervised feature learning with exemplar convolutional neural networks." TPAMI 2015
> [2] R. Zhang et al. "Colorful image colorization." ECCV 2016.
> [3] P. Bojanowski et al. "Unsupervised learning by predicting noise." ICML 2017.
> [4] D. Pathak et al. “Context Encoders: Feature Learning by Inpainting". CVPR 2016.
> [5] M. Noroozi "Unsupervised learning for visual representations by solving jigsaw puzzles." ECCV 2016

---

### Official Review · AnonReviewer2 · 2019-10-24
**Official Blind Review #2**

**Rating:** 1

**Review:**

This paper explores self-supervised learning in the low-data regime, comparing results to self-supervised learning on larger datasets.  BiGAN, RotNet, and DeepCluster serve as the reference self-supervised methods.  It argues that early layers of a convolutional neural network can be effectively learned from a single source image, with data augmentation.  A performance gap exists for deeper layers, suggesting that larger datasets are required for self-supervised learning of useful filters in deeper network layers.

I believe the primary claim of this paper is neither surprising nor novel.  The long history of successful hand-designed descriptors in computer vision, such as SIFT [Lowe, 1999] and HOG [Dalal and Triggs, 2005], suggest that one can design (with no data at all) features reminiscent of those learned in the first couple layers of a convolutional neural network (local image gradients, followed by characterization of those gradients over larger local windows).

More importantly, it is already well established that it is possible to learn, from only a few images, filter sets that resemble the early layers of filters learned by CNNs.  This paper fails to account for a vast amount of literature on modeling natural images that predates the post-AlexNet deep-learning era.

For example, see the following paper (over 5600 citations according to Google scholar):

[1] Bruno A. Olshausen and David J. Field.  Emergence of simple-cell receptive field properties by learning a sparse code for natural images.  Nature, 1996.

Figure 4 of [1] shows results for learning 16x16 filters using "ten 512x512 images of natural scenes".  Compare to the conv1 filters in Figure 2 of the paper under review.  This 1996 paper clearly established that it is possible to learn such filters from a small number of images.  There is long history of sparse coding and dictionary learning techniques, including multilayer representations, that follows from the early work of [1].  The paper should at minimum engage with this extensive history, and, in light of it, explain whether its claims are actually novel.

**Experience Assessment:**

I have published in this field for several years.

**Review Assessment: Checking Correctness Of Derivations And Theory:**

N/A

**Review Assessment: Checking Correctness Of Experiments:**

I carefully checked the experiments.

**Review Assessment: Thoroughness In Paper Reading:**

I read the paper thoroughly.

---

> ### Author Response · Authors · 2019-11-07
> **Response to Review #2**
>
> We hope that the reviewer will change his opinion once we clarify the goal of our paper and explain how it relates to prior work, as we believe we are fundamentally on the same page.
>
> We are well aware of SIFT, HOG, the results of Olshausen and Field on learning image filters from a few example images (some of us are sufficiently old to have implemented all such methods from scratch as grad students!) and no annotations, as well as Mallat’s Scattering nets [1]. In fact, we discuss and evaluate Oyallon’s 2017 implementation [2] of this at page 5 and table 2 in the paper.
>
> However, the existence of these methods does not detract from the message of this paper. Our goal is to provide “critical analysis” of current self-supervision methods because these *specific* tools are now very heavily researched.  Our paper sends a cautionary message: current self-supervised learning techniques cannot improve on what can be obtained from a single image plus transformations for early layers in a network, and only improves in a limited manner for deeper layers, despite ingesting millions of images (which is touted as their key advantage). In particular, the claims are not limited to the first few layers as we show that one image recovers two thirds of the performance of deeper layers as well. This message, which is a partially negative result, stands on its own, regardless of whether good low-level features can be obtained in some other ways (e.g. manually) and, we hope the reviewer will agree, should be known by the community.
>
> Nevertheless, we also agree with the reviewer that it is interesting to put these findings in a broader context, so we are happy to expand the discussion of prior feature learning/design work further. However, please note that none of this literature makes our specific findings on the limits of self-supervision obvious. Furthermore, although this is a little besides the point, in the paper we do show in Table 2 that scattering transforms works as well as conv1, but that from conv2 onwards self-supervision on a single image does better, so even the claim that handcrafted features are equivalent to the first few layers in deep networks is not proven. Also, the fact that Olshausens’s filters resemble conv1 does not mean that they are equivalent to conv1 in recognition performance.
>
> —
> [1] J. Bruna and S. Mallat. "Invariant scattering convolution networks." TPAMI 2013
> [2] E. Oyallon, et al. "Scaling the scattering transform: Deep hybrid networks." ICCV 2017

---

### Official Review · AnonReviewer1 · 2019-10-26
**Official Blind Review #1**

**Rating:** 6

**Review:**

Update 11/21
With the additional experiments (testing a new image, testing fine-tuning of hand-crafted features), additions to related work, and clarifications, I am happy to raise my score to accept. Overall, I think this paper is a nice sanity check on recent self-supervision methods. In the future, I am quite curious about how these mono-image learned features would fare on more complex downstream tasks (e.g., segmentation, keypoint detection) which necessarily rely less on texture.

Summary
This paper seeks to understand the role of the *number of training examples* in self-supervised learning with images. The usefulness of the learned features is evaluated with linear probes at each layer for either ImageNet or CiFAR image classification. Empirically, they find that a single image along with heavy data augmentation suffices for learning the first 2-3 layers of convolutional weights, while later layers improve with more self-supervised training images. The result holds for three state-of-the-art self-supervised methods, tested with two single-image training examples.

In my view, learning without labels is an important problem, and it is interesting what can be learned from a single image and simple data augmentation strategies.

Comments / Questions
It seems to me that for completeness, Table 4 should include the result of training a supervised network on top of random conv1/2 and Scattering network features, because this experiment is actually testing what we want - performance of the features when fine-tuned for a downstream task. So for example, even if a linear classifier on top of Scattering features does poorly, if downstream fine-tuning results in the same performance as another pre-training method, then Scattering is a perfectly fine approach for initial features. Could the authors please either correct this logic or provide the experiments?
Further, it seems that the results in Table 4 might be a bit obscured by the size of the downstream task dataset. I wonder if the learned features require fewer fully supervised images to obtain the same performance on the downstream task?
Can the authors clarify how the neural style transfer experiment is performed? The method from Gatys et al. requires features from different layers of the feature hierarchy, including deeper layers. Are all these features taken directly from the self-supervised network or is it fine-tuned in some way?
While I appreciate the computational burden of testing more images, it does feel that Image A and B are quite cherry-picked in being very visually diverse. Because of this, it seems like a precise answer to what makes a good single training image remains unknown. I wonder how feasible it is to find a proxy metric that corresponds to the performance on downstream tasks which is expensive to compute. It might be interesting to try to generate synthetic images (or modify real ones) that are good for this purpose and observe their properties.
I disagree with the claim of practicality in the introduction (page 2, top). While training on one image does reduce the burden of number of images, the computational burden remains the same. And as mentioned above, it doesn’t seem likely that *any* image would work for this method. Finally, more images are needed to learn the deeper layers for the downstream task anyway.

The paper is well-written and clear.


**Experience Assessment:**

I have read many papers in this area.

**Review Assessment: Checking Correctness Of Derivations And Theory:**

N/A

**Review Assessment: Checking Correctness Of Experiments:**

I carefully checked the experiments.

**Review Assessment: Thoroughness In Paper Reading:**

I read the paper thoroughly.

---

> ### Author Response · Authors · 2019-11-07
> **Response to Review #1**
>
> We thank the reviewer for their time and detailed reading of the paper. In the following, we address each of the reviewers comments:
>
> > Table 4 should include the result of training a supervised network on top of random conv1/2 and Scattering network features […] Scattering is a perfectly fine approach for initial features.
>
> Our aim is to investigate the “power” (or lack thereof) of current self-supervision techniques when applied to standard deep network models. This is of interest because self-supervision is a hot topic of research.
>
> Finding whether e.g. the Scattering Transform can replace the first few layers of a network is interesting, for example to know if handcrafted features can also do as well as (self) supervision for the first few layers, but not susbtitutive of our core investigation (furthermore, we also look ad deeper layers, where these features are unlikely to be competitive). Still, such an experiment can help put our findings in context. This is why we do include them in Table 2 of the paper, where we show that scattering is not quite as good as even single-image self-supervision.
>
> We do think that the suggestion of finetuning/retraining the rest of the model is also interesting after replacing the first few layers is also interesting. Still, we think that this can complement but not replace linear probing as the latter is a more direct way of finding what the probed layers can do. For instance, it is likely possible to learn a good network even by replacing the first layer with the identity function — it is just a slightly less deep model.
>
> For these two reasons, we are running the requested experiments and we hope to be able to update Table 4 in the following days.
>
> > Can the authors clarify how the neural style transfer experiment is performed?
>
> Indeed, the method by Gatys et al. uses deeper layers as well, which we also use — straight from the self-supervised method, without fine-tuning or anything else. We will update the paper with these details.
>
> > While I appreciate the computational burden of testing more images, it does feel that Image A and B are quite cherry-picked in being very visually diverse. [...]  It might be interesting to try to generate synthetic images (or modify real ones) that are good for this purpose and observe their properties.
>
> Thank you, finding the best single training image, or finding useful synthetic images, are both very interesting ideas. While we are happy to consider doing so as a next step, it is next to impossible to do so in time for the rebuttal (we do not have access to thousands of GPUs).
>
> Nevertheless, we would argue that the paper stands on by making some interesting observation on the ability of self-supervision to extract useful information from more than one (or few) images, and by investigating the role of data augmentation in this process. We hope that the reviewer will agree that the community will be interested in hearing about these findings.
>
> > I disagree with the claim of practicality in the introduction (page 2, top). While training on one image does reduce the burden of number of images, the computational burden remains the same.
>
> Our intention wasn’t to say that we can save compute time, but data collection effort (which is also a practical issue in some applications). Nevertheless, we agree that our findings have mostly a theoretical value, so we have adjusted the wording to reflect that.
>
> > And as mentioned above, it doesn’t seem likely that *any* image would work for this method.
>
> It is true that we did not quite prove that, so we have reworded the text to tone down this claim.
>
> To be a bit more specific, obviously a blank image would not work, and textureless images would probably not work well either. However, we did use in the paper the first two images we manually selected from Google Image Search (while we did select images with some texture, they have not been otherwise been optimized for good performance in our evaluation). Thus, we think that it is extremely likely that many other images would work just as well.
>
> > Finally, more images are needed to learn the deeper layers for the downstream task anyway.
>
> True, but even for deeper layers a single image achieves two thirds of the performance that self-supervision can squeeze out of a million images, which we think is interesting.

---

### Author Response · Authors · 2019-11-15
**Final paper update**

We have updated our paper with the following main changes:

* As suggested by R2, we adjusted the message of the paper to more accurately reflect the critical results of this paper and how they relate to previous hand-designed feature learning methods (esp. Sec. 2).
* As requested by R3, we have provided an additional experiment on training on a single, less crowded, image with DeepCluster (in Sec. 4.3) and observe that it can still achieve almost the same performance as with the other photographic image.
* We have also provided freeze-and-retrain experiments for the scattering transform and our single image trained networks (Tab. 4) and find that while the scattering transform does outperform random conv1-conv2, our CNNs trained self-supervisedly with one image still yield better performance.
* Incorporated several smaller clarifications requested by the reviewers.
* Due to the short rebuttal period, the experiment on training a ResNet has not yet finished evaluating and we will provide the finished results in the camera-ready version.

---

### Decision · Program_Chairs · 2019-12-19

**Decision:**

Accept (Poster)

**Comment:**

This paper studies the effectiveness of self-supervised approaches by characterising how much information they can extract from a given dataset of images on a per-layer basis. Based on an empirical evaluation of RotNet, BiGAN, and DeepCluster, the authors argue that the early layers of CNNs can be effectively learned from a single image coupled with strong data augmentation. Secondly, the authors also provide some empirical evidence that supervision might still necessary to learn the deeper layers (even in the presence of millions of images for self-supervision).
Overall, the reviews agree that the paper is well written and timely given the growing popularity of self-supervised methods. Given that most of the issues raised by the reviewers were adequately addressed in the rebuttal, I will recommend acceptance. We ask the authors to include additional experiments requested by the reviewers (they are valuable even if the conclusions are not perfectly aligned with the main message).